# Patient capital and green total factor productivity: Evidence from Chinese listed companies

**Yue Li[1], Xing Huang ORCID[2]\*, Huanqi Luo[2]**

**1** School of Economics, Shenzhen Polytechnic University, Shenzhen, Guangdong, China, **2** School of Economics, Minzu University of China, Beijing, China

☯ These authors contributed equally to this work.
\* 24400039@muc.edu.cn

## Abstract

This study applies Fisher's investment framework to investigate how patient capital affects firms' green total factor productivity (GTFP). Using panel data from Chinese listed companies over the period 2008–2023, we measure firm-level GTFP by employing a non-radial SBM directional distance function combined with the Malmquist–Luenberger productivity index. Our analysis, based on two-way fixed-effects models and instrumental variable regressions, reveals that an increase in patient capital significantly enhances firms' green total factor productivity. Mechanism analysis indicates that this improvement arises from reduced financing costs as well as increased green R&D investment. Furthermore, the positive impact is particularly pronounced among small and medium-sized enterprises, non-state-owned firms, and pollution-intensive industries. The findings suggest that encouraging the transition from short-term speculative capital toward long-term patient capital can effectively improve firms' environmental efficiency. Thus, policy efforts should be directed towards expanding the supply of patient capital to promote corporate sustainability and accelerate the low-carbon economic transition without compromising economic efficiency.

## Introduction

Under China's dual-carbon goals and high-quality development agenda, facilitating enterprises' green transition has become critical. Green total factor productivity (GTFP), reflecting comprehensive production efficiency under environmental constraints, serves as a key indicator of firms' sustainable performance. Yet, investments in green innovation often entail substantial upfront costs and long payback periods, posing challenges for short-term-oriented finance. Thus, patient capital, characterized by long-term investment horizons and risk tolerance, emerges as a promising solution to these green financing challenges. These investments entail large capital requirements and long payback periods, making it difficult for "short-sighted" or

**Data availability statement:** All relevant data are available in the Dryad Digital Repository at https://doi.org/10.5061/dryad.gtht76j0j.

**Funding:** This research was supported by the Ministry of Education of China Humanities and Social Sciences Youth Fund Project "Refining Local Legislation on Social Credit: An Examination of Fifteen Local Regulations" (Grant No. 21YJC820023). The funders had no role in study design, data collection and analysis, decision to publish, or preparation of the manuscript.

**Competing interests:** The authors have declared that no competing interests exist.

"quick-profit" capital supply to satisfy the long-term demands of a green transition. However, existing research predominantly explores patient capital's impacts on conventional corporate outcomes, such as innovation or economic performance, with limited attention to environmental performance at the firm level. Moreover, much literature concentrates on macro-level analyses, neglecting micro-level empirical evidence on how patient capital concretely fosters firms' green transitions. This gap motivates two core research questions: (1) Can patient capital effectively enhance corporate-level GTFP? (2) If so, through which specific mechanisms does patient capital affect firms' green performance?

Against this backdrop, patient capital—characterized by a long-term investment horizon and risk-sharing mechanisms—has steadily emerged as an important break-through in addressing green financing challenges [1]. In contrast to conventional, short-term capital focused on financial metrics, patient capital emphasizes fostering a firm's environmental competitiveness through sustained, long-horizon investment. Its capital cycle dovetails well with the iterative nature of green innovation [2]. At the same time, patient capital can partially alleviate firms' short-term profitability pressures and liquidity risks, offering sustained funding for projects like environmental technology R&D and facilities upgrades that yield returns over a longer horizon [3]. By doing so, patient capital encourages firms to pursue more forward-looking investments in energy conservation and pollution control. However, existing theoretical and empirical studies have not fully examined the concrete role that patient capital plays in facilitating green transitions or its actual impact on corporate GTFP. Motivated by this gap, the present paper centers on whether patient capital can enhance firms' green total factor productivity, aiming to provide new insights for China's enterprise-level green transformation and high-quality development by examining micro-level financing structures and green investment strategies.

Previous research on patient capital has primarily addressed its economic and social implications, highlighting its influence on corporate innovation, long-term development, and social value creation. In economic terms, patient capital can effectively mitigate financing constraints on long-cycle projects and spur innovation [4], provide greater tolerance for R&D failures [5], contribute to deeper involvement in commercializing technological outcomes [6], and help firms undertake cross-cycle strategic planning to improve long-run competitiveness. Regarding social impact, patient capital places more emphasis on social responsibility and sustainability, thus encouraging enterprises to enhance their environmental performance, refine corporate governance, and strengthen social reputation [7], as well as raise long-term value creation through broader social responsibility commitments [8].

Meanwhile, the field of environmental economics has established a GTFP-centric measurement framework. Directional distance functions enable the inclusion of pollution emissions into productivity analyses [9], while the development of a non-radial slack-based measure (SBM) further refines performance evaluations under dual constraints of economy and environment [10]. The Porter hypothesis posits that environmental regulation can stimulate technological innovation and achieve a win–win scenario for economy and environment [11,12], while research on biased

green technological progress emphasizes the pivotal role of green innovation in fostering long-term economic growth [13]. In China's context, green finance channels capital into low-carbon sectors, fueling industrial restructuring [14,15], and environmental regulation aligns ecological efficiency with economic growth via internalizing pollution costs [12]. Firm-level evidence additionally shows threshold effects in environmental investment [16], that green technology innovation substantially reduces firms' pollution abatement costs [17], and that enterprises with robust ESG performance are more likely to obtain long-term capital support [18]. Furthermore, the mismatch between the lengthy cycle of clean technology R&D and the short-term nature of traditional finance has underscored the need for cross-period financing and risk-sharing models [19].In response, recent studies have emphasized patient capital's role in facilitating ESG performance and green innovation, highlighting its potential for driving sustainable corporate growth [1,20]. At the same time, green innovation research underscores the crucial influence of environmental policies and digital transformation on firms' productivity [21,22].

However, existing research exhibits two notable deficiency. First, prior theoretical frameworks rarely integrate patient capital explicitly with firm-level green productivity. Most studies have primarily focused on how patient capital influences conventional economic outcomes, such as corporate R&D and general innovation, with limited consideration of environmental implications. Second, existing empirical research has predominantly emphasized macro-level analyses of environmental regulation and green finance, leaving firm-level evidence underexplored—particularly regarding how patient capital specifically fosters firms' green transitions and enhances environmental performance.

This paper offers three key contributions. First, theoretically, we extend Fisher's two-period investment model to clarify how patient capital, characterized by long-term orientation and higher risk tolerance, enhances firms' green total factor productivity (GTFP). Second, empirically, we identify two core mechanisms—reduced financing costs and increased green R&D investment—through which patient capital improves firms' green performance. Third, through heterogeneity analyses, we demonstrate stronger effects for SMEs, non–state-owned enterprises, and pollution-intensive sectors, providing micro-level evidence useful for targeted policy-making in green finance and sustainable firm transition.

We develop a theoretical framework that links patient capital to firms' green transition and derive testable implications. Guided by these hypotheses, we assemble firm-level data and construct variables that capture both capital characteristics and green productivity. The empirical analysis employs a high-dimensional fixed effects design to establish causal effects, complemented by a series of robustness and endogeneity checks. We then investigate the mechanisms through which patient capital shapes green outcomes and assess heterogeneity across firm and regional contexts. The paper concludes by drawing policy implications for sustainable finance and outlining avenues for future research.

## Policy background and theoretical mechanism

### Policy background

Over the past three decades, China's capital market has gradually transitioned from a predominantly short-term financing structure toward a more diversified and long-term-oriented investment and financing system. Recently, with the progressive advancement of green-finance policies and the clear regulatory impetus provided by the national "dual-carbon" goals (carbon peaking and carbon neutrality), the external financing environment and policy directions faced by enterprises have changed significantly. Regulators have actively encouraged the cultivation and expansion of long-term, patient capital, prompting investors to increasingly emphasize environmental, social, and governance (ESG) performance and support corporate green innovation through sustained investment horizons and risk-sharing mechanisms. Thus, the formation, allocation, and effectiveness of patient capital have been shaped profoundly by these external institutional conditions, laying a solid foundation for our analysis of the relationship between patient capital and firms' green total factor productivity.

### Theoretical mechanism

**The direct effect of patient capital on firms' green total factor productivity.** Drawing on Fisher's (1930) two-period investment decision model, this paper examines how patient capital influences a firm's green total factor productivity

(GTFP) [23]. Concretely, in the first period, the firm chooses its financing structure-that is, the proportion of patient capital versus short-sighted capital—and makes corresponding investments. In the second period, it realizes actual output, undertakes environmental obligations, and pays relevant capital costs. Let the firm's total financing need be ; the fraction of patient capital be $\theta$; and the fraction of short-sighted capital be $1 - \theta$. Suppose the interest rates for patient capital and short-sighted capital are $r_p$ and $r_s$, respectively, with $r_p < r_s$. The firm's overall financing cost $FC(\theta)$ can thus be expressed as:

$$FC(\theta) = \theta r_p + (1 - \theta)r_s, \; r_p < r_s \tag{1}$$

Since $r_p < r_s$, it follows that

$$\frac{dFC(\theta)}{d\theta} = r_p - r_s < 0 \tag{2}$$

indicating that a higher proportion of patient capital lowers the firm's overall financing cost. This cost reduction improves the feasibility of larger-scale or longer-horizon environmental investments. Accordingly, the firm's green investment scale (GI) depends on $\theta$, when $\theta$ increases, the firm gains access to lower-cost, longer-term funds, eases short-term debt pressures, and is thereby more likely to invest substantially in environmental technology R&D and facility upgrades. Consequently, GI is positively correlated with $\theta$, supporting the notion that a rising share of patient capital enables the firm to commit more resources to environmental objectives, ultimately fostering an increase in GTFP.

Following the logic above, let a firm's green investment scale be determined by:

$$GI(\theta) = \gamma + \delta\theta, \; \delta > 0, \gamma \geq 0 \tag{3}$$

where $\gamma$ denotes the baseline level of green investment in the firm, and $\delta$ represents the marginal increase in green investment as the proportion of patient capital $\theta$ rises. Provided $\delta > 0$, we have $\frac{dGI(\theta)}{d\theta} = \delta > 0$, implying that an increase in patient capital share makes it easier for the firm to channel more resources into environmental projects.

In the second period, the firm's green total factor productivity GTFP depends on both its green investment (GI) and its overall financing cost (FC). Green investment exerts a positive influence on GTFP, while higher financing costs pose a financial burden on the firm's environmental and R&D commitments [24]. Drawing on the Cobb and Douglas (1928) style of multiplicative production functions [25,26], we define:

$$GTFP = G(GI(\theta), FC(\theta)) = (GI(\theta))^{\alpha}(FC(\theta))^{-\beta}, \; \alpha > 0, \beta > 0 \tag{4}$$

This specification captures how an increase in the scale of green investment improves production efficiency, whereas lower financing costs similarly free up funds for environmental technology upgrades and innovations, thereby promoting GTFP. Substituting

$$GI(\theta) = \gamma + \delta\theta \;\; \text{and} \;\; FC(\theta) = \theta r_p + (1 - \theta)r_s \tag{5}$$

into the above function yields:

$$GTFP(\theta) = H(\theta) = [\gamma + \delta\theta]^{\alpha} [\theta r_p + (1 - \theta)r_s]^{-\beta} \tag{6}$$

Our goal is to examine the sign of $\frac{dH(\theta)}{d\theta}$. To simplify the analysis, we first take the natural logarithm of $H(\theta)$:

$$\ln H(\theta) = \alpha\ln[\gamma + \delta\theta] - \beta\ln[\theta r_p + (1 - \theta)r_s] \tag{7}$$

Differentiating (5) with respect to θ yields:

$$\frac{d\ln H(\theta)}{d\theta} = \alpha\frac{\delta}{\gamma + \delta\theta} - \beta\frac{d\left[\theta r_p + (1-\theta)r_s\right]/d\theta}{\theta r_p + (1-\theta)r_s} \tag{8}$$

Since

$$\frac{d}{d\theta}\left[\theta r_p + (1-\theta)r_s\right] = r_p - r_s \tag{9}$$

we have:

$$\frac{d\ln H(\theta)}{d\theta} = \alpha\frac{\delta}{\gamma + \delta\theta} - \beta\frac{r_p - r_s}{\theta r_p + (1-\theta)r_s} \tag{10}$$

Recalling $r_p < r_s$, we know $r_p - r_s < 0$. The first term $\alpha(\delta/[\gamma + \delta\theta])$ is strictly positive, and the second term $-\beta\left((r_p - r_s)/[\theta r_p + (1-\theta)r_s]\right)$ is also strictly positive (because $r_p - r_s < 0$ and there is a leading minus sign). Consequently,

$$\frac{d\ln H(\theta)}{d\theta} > 0 \Rightarrow \frac{dH(\theta)}{d\theta} > 0 \tag{11}$$

This implies that a higher proportion of patient capital θ unambiguously raises the firm's green total factor productivity $H(\theta)$. Intuitively, a rise in θ not only lowers overall financing costs but also encourages larger green investments, forming a mutually reinforcing effect that leads to an increase in GTFP.

Following the log-differentiation of $H(\theta)$ and reverting to the original function's derivative, we have:

$$\frac{dH(\theta)}{d\theta} = H(\theta)\frac{d\ln H(\theta)}{d\theta} \tag{12}$$

By substituting the expressions derived previously, it gives:

$$\frac{dH(\theta)}{d\theta} = H(\theta)\left[\alpha\frac{\delta}{\gamma + \delta\theta} - \beta\frac{r_p - r_s}{\theta r_p + (1-\theta)r_s}\right] \tag{13}$$

In analyzing the sign of this derivative, recall that $\gamma + \delta\theta > 0, \theta r_p + (1-\theta)r_s > 0$, and $\delta > 0, \alpha > 0, \beta > 0$, alongside $r_p < r_s \Rightarrow r_p - r_s < 0$. Therefore,

$$\alpha\frac{\delta}{\gamma + \delta\theta} > 0, \quad -\beta\frac{r_p - r_s}{\theta r_p + (1-\theta)r_s} > 0 \tag{14}$$

which ensures

$$\frac{dH(\theta)}{d\theta} > 0 \tag{15}$$

Hence, an increment in θ (the proportion of patient capital) unambiguously raises $H(\theta)$, the firm's green total factor productivity. Formally:

$$\frac{dGTFP(\theta)}{d\theta} > 0 \tag{16}$$

Based on these theoretical derivations, we propose the following hypothesis:

H1: Patient capital can enhance firms' green total factor productivity.

**Indirect effect of patient capital on firms' green total factor productivity.** From the perspective of financing constraints theory, if a firm's external financing channels are limited or the cost of capital is high, it often cannot devote substantial resources to environmentally oriented projects that carry long payback horizons and significant technical risks. Since the R&D of environmental technologies and the upgrading of energy-saving equipment typically involve considerable costs over extended timeframes, short-term capital may withdraw support or press for reductions in environmental budgets when results do not appear promptly [27], causing firms to abandon ongoing green innovation or cease core eco-friendly process improvements [28]. Consequently, many enterprises face a funding bottleneck when promoting green transitions, making it difficult to achieve effective progress in areas such as energy efficiency and pollution reduction.

Unlike conventional short-term capital, patient capital has a longer investment horizon and a higher tolerance for risk. Therefore, it can consistently support environmentally focused projects even when tangible returns have yet to materialize [28], mitigating the likelihood that short-term profit pressures or inadequate cash flow will force the firm to discontinue environmental R&D. Through this relatively flexible financing model, firms can more boldly expand their efforts in green technology iteration and equipment upgrades [29], without worrying that short-term profit fluctuations will trigger capital withdrawal or stringent constraints from external investors. This latitude facilitates continuous refinement of green technologies and processes, thereby reducing the non-desired outputs in production—such as emissions, waste, or resource depletion and ultimately increasing overall production efficiency [30]. Based on the above theoretical analysis, we propose the following hypothesis:

H2: Patient capital enhances firms' green total factor productivity by alleviating financing constraints.

From a theoretical standpoint, firms pursuing green transitions often confront the dual challenge of substantial R&D expenditures and prolonged payback horizons. By adhering to a long-term investment philosophy and a more tolerant attitude toward short-term returns, patient capital can substantially alleviate both the financial and risk-related pressures associated with green R&D. Drawing on Porter's perspective of competitive advantage, environmentally oriented innovation typically demands repeated technical experimentation, iterative process improvements, and sustained investment. If a firm relies solely on capital seeking rapid payoffs, it may abandon critical environmental technologies whenever it encounters developmental bottlenecks or temporary losses [31]. In contrast, patient capital emphasizes a longer capital supply horizon and provides more stable, ongoing support for a firm's green R&D endeavors, sparing the firm from constant pressure to deliver immediate profits or results. Given the intrinsic uncertainties and relative immaturity of some environmental technologies, firms that perpetually fear capital or market backlash are inclined to cut short or discontinue R&D well before such projects can yield tangible benefits. Patient capital [32], however, endows management with greater autonomy and scope for exploration [33], allowing deeper engagement in low-carbon processes and energy conservation technologies. From the resource-based view, the cumulative process of R&D investment strengthens a firm's specialized capabilities in green technology [34], fostering higher levels of production efficiency and environmental performance. The infusion of long-term capital enables management to maintain consistent inputs into environmental research and development, paving an iterative path of technological refinement that eventually raises a firm's overall green total factor productivity [35]. In this sense, once a firm secures patient capital, it gains a more sustainable funding environment for green technological innovation and stands to reap enhanced returns in both emissions reduction and ecological benefits. In line with the above theoretical analysis, we propose the following hypothesis:

H3: Patient capital enhances firms' green total factor productivity by increasing their green R&D investments.

## Variable selection and model specification

### Model specification

To investigate whether and how patient capital (PatCap) influences firms' green total factor productivity (GTFP), this paper adopts a two-way fixed effects model:

$$\text{GTFP}_{i,t} = \alpha_0 + \alpha_1 \text{PatCap}_{i,t} + \alpha_2 \text{Controls}_{i,t} + \lambda_i + \gamma_t + \varepsilon_{i,t} \tag{17}$$

where $\text{GTFP}_{i,t}$ denotes the green total factor productivity of firm i in year t. The variable $\text{PatCap}_{i,t}$ measures the extent of patient capital available to firm i. The term $\text{Controls}_{i,t}$ represents a set of control variables (such as firm size, leverage, industry competition intensity, etc.). Meanwhile, $\lambda_i$ and $\gamma_t$ capture firm fixed effects and time fixed effects, respectively, which help to control for unobserved heterogeneity across firms and macro-level temporal trends. Lastly, $\varepsilon_{i,t}$ is the stochastic error term. If $\alpha_1$ is significantly positive, it implies that patient capital can effectively enhance GTFP, thereby supporting the core hypothesis of this study. Conversely, if $\alpha_1$ is insignificant or significantly negative, it indicates that the relationship between patient capital and GTFP is weaker or contrary to expectations.

### Variable specification

**Green total factor productivity.** Compared with the conventional total factor productivity (TFP) measure, green total factor productivity (GTFP) explicitly incorporates resource and energy consumption as well as environmental pollution emissions (i.e., undesirable outputs), enabling a more objective reflection of firms' actual production efficiency under environmental constraints. Following prior studies [36], this paper employs the non-radial, non-oriented SBM directional distance function (SBM-DDF) in conjunction with the Malmquist-Luenberger (ML) index to gauge each sample firm's GTFP. Concretely, based on the SBM-DDF model, each firm is treated as a decisionmaking unit ($\text{DMU}_j, j = 1,2,\ldots,q$). Each DMU's production process involves three categories of variables: input (I), desirable output (O), and undesirable output (P). Specifically, I represents the production factors a firm invests (such as capital, energy, and labor); O corresponds to the firm's normal or desirable output (e.g., operating revenue); and P comprises pollutant emissions or other negative externalities (e.g., wastewater, exhaust gas).

Let the input matrix be $I = [i_1, i_2, \ldots, i_q] \in R^{a \times q}$, the desirable output matrix be $O = [o_1, o_2, \ldots, o_q] \in R^{b \times q}$, and the undesirable output matrix be $P = [p_1, p_2, \ldots, p_q] \in R^{c \times q}$. The SBM-DDF framework solves a linear optimization problem that simultaneously adjusts inputs, expands desirable outputs, and contracts undesirable outputs, thus capturing a firm's efficiency under both economic and environmental constraints. By solving for the direction distance function, we then integrate the Malmquist-Luenberger index to track dynamic changes in GTFP across periods, reflecting both efficiency shifts and technological progress in green production. This combined measurement approach, grounded in the SBM-DDF model and the ML index, offers a comprehensive view of the green production efficiency for each sample firm. Model (13) below formulates the non-radial, non-oriented SBM directional distance function:

$$\min \phi = \frac{1 - \frac{1}{A} \sum_{a=1}^{A} \frac{t_a^i}{i_{a0}}}{1 + \frac{1}{B+C} \left( \sum_{b=1}^{B} \frac{t_b^o}{o_{b0}} + \sum_{c=1}^{C} \frac{t_c^p}{p_{c0}} \right)} \tag{18}$$

subject to the following constraints:

$$\begin{aligned}
\sum_{j=1}^{q} w_j i_{aj} + t_a^i &= i_{a0}, & a &= 1,2,\ldots,A \\
\sum_{j=1}^{q} w_j o_{bj} - t_b^o &= o_{b0}, & b &= 1,2,\ldots,B \\
\sum_{j=1}^{q} w_j p_{cj} + t_c^p &= p_{c0}, & c &= 1,2,\ldots,C, \\
\sum_{j=1}^{q} w_j &= 1, w_j, t_a^i, t_b^o, t_c^p \geq 0,
\end{aligned} \tag{19}$$

where $\varphi$ represents the static efficiency measure of a firm's inputs and outputs; $t$ denotes the slack variables for inputs, desirable outputs, and undesirable outputs, respectively; $w_j$ is the weight of each decision-making unit (DMU) on the efficiency frontier. Furthermore, once the efficiency value has been obtained, the model allows decomposition of production inefficiency into two components-input inefficiency (IE) and output inefficiency (OE):

$$IE = \frac{1}{A} \sum_{a=1}^{A} \frac{t_a^i}{i_{a0}}, \quad OE = \frac{1}{B+C} \left( \sum_{b=1}^{B} \frac{t_b^o}{o_{b0}} + \sum_{c=1}^{C} \frac{t_c^p}{p_{c0}} \right) \tag{20,21}$$

After obtaining the directional distance function from the above model, we combine it with the Malmquist-Luenberger (ML) index to compute the dynamic evolution of firms' green total factor productivity (GTFP). This integrated approach captures both static efficiency and technological changes over time, thereby offering a more comprehensive perspective on a firm's green production performance.

After obtaining the directional distance function from the above SBM-DDF model, we incorporate the Malmquist-Luenberger (ML) index to evaluate each firm's dynamic changes in green total factor productivity (GTFP). The ML index from period $s$ to $s + 1$ is defined as:

$$ML\_GTFP_s^{s+1} = \left[ \frac{1 + D_s^0\left(i^s, o^s, p^s; g^{s+1}\right)}{1 + D_s^0\left(i^{s+1}, o^{s+1}, p^{s+1}; g^{s+1}\right)} \times \frac{1 + D_{s+1}^0\left(i^s, o^s, p^s; g^{s+1}\right)}{1 + D_{s+1}^0\left(i^{s+1}, o^{s+1}, p^{s+1}; g^{s+1}\right)} \right]^{\frac{1}{2}} \tag{22}$$

where $D_s^0(\cdot)$ and $D_{s+1}^0(\cdot)$ are the directional distance functions in periods $s$ and $s + 1$, respectively, and $(i,o,p)$ represent inputs, desirable outputs, and undesirable outputs for the firm in each period. In addition, this ML index can be decomposed into an efficiency change (EC) component and a technical progress (TC) component:

$$ML_s^{s+1} = EC \times TC \tag{23}$$

A value of $ML_s^{s+1} > 1$ indicates that the firm's green total factor productivity has improved from period $s$ to $s + 1$. Conversely, a value less than 1 signifies that the firm's GTFP declined over the examined interval. By integrating the SBM-DDF approach with the ML index, this paper obtains a more accurate and comprehensive measure of a firm's green production efficiency, providing essential empirical evidence for subsequent analysis on the relationship between patient capital and GTFP.

Building on the SBM-DDF framework and the Malmquist-Luenberger (ML) index, this paper measures firm-level green total factor productivity (GTFP) for Chinese A-share listed companies, following the approach in [37]. The variables are organized along three dimensions: (1) input variables; (2) desirable output; and (3) undesirable output.

Input Variables: Capital Investment: Using a perpetual inventory method, we estimate each firm's capital stock. Specifically, $K_{i,t} = K_{i,t-1}(1 - \delta) + I_{i,t}$, where $\delta = 9.6\%$ is the depreciation rate and $I_{i,t}$ represents the firm's fixed-asset investment expenditure in year $t$. Energy Inputs: Measured by the annual energy consumption disclosed in company reports. Labor: Approximated by the number of employees recorded in annual reports.

Desirable Output: Following standard practice for listed firms, we adopt operating revenue as the proxy for each firm's desirable output indicator, adjusting values to constant prices using 2008 as the base year and deflating with an appropriate GDP deflator.

Undesirable Output: We select industrial wastewater, industrial $SO_2$, and industrial smoke/dust emissions to capture the environmental negative externalities generated during each firm's production and operation processes.

These GTFP measures jointly reflect resource inputs and environmental influences on economic production. Capital, energy, and labor serve as core production factors, while energy inputs further illustrate the resource-use efficiency within

a firm's economic activities. Industrial pollution indicators reflect the degree of environmental externalities arising from production. By comparing across industries and firms, we can obtain deeper insights into how patient capital affects firms' green productivity enhancements. Based on the above inputs and outputs, this paper utilizes MaxDEA software to calculate each firm's GTFP from 2008 to 2023. The ML index indicates each year's change in green total factor productivity relative to the previous year. We set 2008 as the base period with a GTFP level of 1, and multiply that level by each year's ML index to derive each firm's annual GTFP value.

**Patient capital.** This paper's core explanatory variable is patient capital, employed to gauge the extent of stable, long-term capital support accessible to a firm. Specifically, the measurement proceeds along two dimensions—debt capital and equity capital. First, following David et al. (2008) and Wen et al. (2011) [4,38], this study defines the long-term debt capital ratio (LDC) as the firm's total long-term borrowing divided by its total liabilities (namely, longterm loans, short-term loans, and bonds payable), reflecting both the stability and term characteristics of its debt capital. Second, in view of potential differences between domestic and foreign institutional investors, we draw on the methods proposed by Niu et al. (2013) and Li et al. (2014) [39,40], selecting the overall institutional shareholding ratio (INST) to capture the long-term nature of equity capital, and further construct a "patient capital stability" index as follows:

$$PatCap_{i,t} = \frac{INST_{i,t}}{STD\left(INST_{i,t-3}, INST_{i,t-2}, INST_{i,t-1}\right)}$$

(24)

Here, $PatCap_{i,t}$ denotes firm i 's shareholding stability from institutional investors in year t, with INST representing the institutional shareholding ratio, and the denominator being the standard deviation of that ratio over the preceding three years. A higher PATI value indicates greater stability of institutional shareholdings over time, implying a higher degree of patient capital support for the firm. Considering that the resulting indicator has a unit dimension, the variable for patient capital is taken in logarithmic form.

**Control variables.** This paper includes the following firm-level and external environmental controls to avoid omitting other potential influencing factors. At the firm level, we incorporate variables such as firm size (measured by the log of total assets or operating revenue) to capture scale effects, board characteristics to reflect corporate governance, firm growth potential (e.g., revenue growth rate) to capture developmental capacity, the shareholding ratio of the largest shareholder to indicate ownership concentration, and metrics like cash flow or leverage ratio to represent financial flexibility and capital structure differences. We also include firm age, reflecting the firm's developmental stage. Externally, we include yearly and industry dummy variables to account for macroeconomic fluctuations, industry competition intensity, and policy or regulatory variations. By incorporating this broad set of control variables, we aim to more accurately identify the effect of patient capital on green total factor productivity. The detailed definitions of each variable are provided in Table 1.

## Sample selection and data sources

This paper initially takes A-share listed firms on the Shanghai and Shenzhen stock exchanges from 2008 to 2023 as the sample. The data-screening steps are as follows: Exclude financial companies to avoid the unique operating traits of the finance industry skewing the results. Remove ST and *ST firms (special treatment labels) to ensure data quality and eliminate firms with abnormal operations. Exclude firms suffering severe missing values in key variables, thereby ensuring data robustness. Apply a 1% Winsorize treatment on continuous variables to lessen the effects of extreme outliers.

After these filtering steps, the final effective sample contains 2002 companies with 32,032 firm-year observations. Most financial and operational data for these listed firms come from the CSMAR and Wind databases; institutional shareholding ratios are drawn from the Wind finance module's ownership holdings; and corporate environmental information (e.g., industrial wastewater, industrial exhaust emissions, and solid waste discharges) is gathered from company annual reports, CSR

**Table 1. Variable definitions.**

| Variables | Definition | Variable types | Unit |
|---|---|---|---|
| GTFP | Green total factor productivity of the firm | Dependent | – |
| LnPatienceCap | Natural log of patient capital | Independent | – |
| LnAsset | Natural log of total assets | Control | – |
| Lev | Leverage (total liabilities/ total assets) | Control | – |
| LnAge | Firm age (number of years from establishment to current) | Control | – |
| LnBoard | Board size (natural log of the number of board directors) | Control | – |
| Growth | Revenue growth rate (current year's operating revenue ÷ previous year) | Control | – |
| Top1Holder | Shareholding ratio of the largest shareholder | Control | – |
| ROE | Return on equity (net income ÷ shareholders' equity) | Control | – |
| CashFlow | Operational cash flow status (cash flow from operations ÷ total assets) | Control | – |

disclosures, environmental disclosures, or government environmental statistics yearbooks. In addition, macroeconomic indicators and price indices are sourced from the National Bureau of Statistics and the China Statistical Yearbook. The data processing and regression analysis are performed in Stata, while the calculation of GTFP values employs MaxDEA. Missing data are imputed by mean interpolation. The principal variables are summarized in descriptive statistics in Table 2.

## Empirical results analysis

### Baseline regression

Building on the aforementioned model, this paper employs a two-way fixed effects regression to estimate the impact of patient capital on firms' green total factor productivity (GTFP). As shown in the results of Table 3, the coefficient on patient capital is significantly positive, indicating that an increase in patient capital can markedly enhance a firm's GTFP. This finding aligns with our hypothesis, suggesting that patient capital, through its long-term horizon and risk tolerance, effectively facilitates enterprises' green transition. Regarding control variables, firm size, board size, firm growth potential, the shareholding ratio of the largest shareholder, and cash flow all exhibit significant positive coefficients, implying that larger firms, those with well-structured boards, stronger growth prospects, better governance arrangements, and more ample cash flow tend to achieve higher green production efficiency. Conversely, leverage and firm age are significantly negative, implying that enterprises with higher financial leverage or a longer operating history demonstrate relatively lower levels of green production efficiency. Consequently, these results support Hypothesis H1.

**Table 2. Descriptive statistics.**

| Variables | N | Mean | SD | Min | P25 | Median | P75 | Max |
|---|---|---|---|---|---|---|---|---|
| GTFP | 32032 | 0.960 | 0.226 | 0.490 | 0.798 | 0.974 | 1.131 | 1.412 |
| LnPatienceCap | 32032 | 1.803 | 1.576 | −1.588 | 0.649 | 1.676 | 2.872 | 5.944 |
| LnAsset | 32032 | 22.346 | 1.405 | 19.144 | 21.383 | 22.193 | 23.189 | 26.386 |
| Lev | 32032 | 0.482 | 0.223 | 0.069 | 0.312 | 0.479 | 0.637 | 1.131 |
| LnAge | 32032 | 2.452 | 0.636 | 1.099 | 2.079 | 2.565 | 2.944 | 3.367 |
| LnBoard | 32032 | 2.147 | 0.199 | 1.609 | 2.079 | 2.197 | 2.197 | 2.708 |
| Growth | 32032 | 0.005 | 0.017 | −0.008 | −0.000 | 0.001 | 0.004 | 0.133 |
| Top1Holder | 32032 | 0.435 | 0.149 | 0.184 | 0.316 | 0.412 | 0.537 | 0.833 |
| ROE | 32032 | −0.248 | 2.018 | −15.030 | −0.096 | −0.020 | 0.065 | 4.764 |
| CashFlow | 32032 | 0.047 | 0.083 | −0.233 | 0.004 | 0.045 | 0.091 | 0.300 |

 

**Table 3. Baseline regression results.**

| Variables | (1) | (2) |
|---|---|---|
| PatienceCap | 0.005*** | 0.009*** |
| | (5.952) | (13.275) |
| LnAsset | | 0.054*** |
| | | (19.050) |
| Lev | | −0.090*** |
| | | (−8.841) |
| LnAge | | −0.041*** |
| | | (−4.892) |
| LnBoard | | 0.052*** |
| | | (5.235) |
| Growth | | 0.124** |
| | | (2.044) |
| Top1Holder | | 0.537*** |
| | | (21.099) |
| ROE | | 0.000 |
| | | (0.868) |
| CashFlow | | 0.026** |
| | | (2.001) |
| _cons | 0.951*** | −0.468*** |
| | (656.610) | (−6.964) |
| Year FE | YES | YES |
| Firm FE | YES | YES |
| N | 32032 | 32032 |
| $R^2$ | 0.792 | 0.843 |

The values in parentheses are t-statistics computed with cluster-robust standard errors; ***, **, and * indicate significance at the 1%, 5%, and 10% levels, respectively. The same applies to all subsequent tables.

## Endogeneity test

To further mitigate potential endogeneity in the patient capital indicator, we employ two categories of instrumental variables and apply a two-stage least squares (2SLS) approach. The first category is the lagged level of patient capital, which is highly correlated with the firm's current patient capital but should not directly affect the firm's current green total factor productivity. The second category is the average patient capital ratio of other firms in the same industry, which is closely linked to the firm's own patient capital in this period yet does not directly enter the firm's green efficiency function, thereby satisfying exogeneity. As reported in Table 4 column (1), these two types of instruments exhibit substantial explanatory power for patient capital, passing the weak-instrument test (F-statistic > 10). Meanwhile, the second-stage regression results in column (2) show that patient capital continues to exert a significantly positive effect on green total factor productivity [41], verifying both the effectiveness and exogeneity of our chosen instruments and corroborating the robustness of our core conclusion.

## Robustness checks

To further confirm the reliability of our empirical findings, this study conducts four key robustness checks. First, we replace the main explanatory variable with the long-term debt capital ratio (LDC), thereby remeasuring patient capital's long-term

**Table 4. Endogeneity test regression results.**

| Variables | (1) | (2) |
|---|---|---|
| L.PatienceCap | 0.042*** | |
| | (19.523) | |
| PatienceCap | | 0.039*** |
| | | (4.755) |
| Kleibergen-Paap rk LM statistic | | 251.596 [0.000] |
| Kleibergen-Paap rk Wald F statistic | | 911.282 {314.648} |
| _cons | −0.722*** | −0.717*** |
| | (−17.558) | (−15.593) |
| Controls | YES | YES |
| Year FE | YES | YES |
| Firm FE | YES | YES |
| N | 30030 | 32032 |
| $R^2$ | 0.514 | 0.515 |

The numeric values in square brackets represent the corresponding p-values of the relevant statistics; the values in curly braces denote the 10% critical values from the Stock–Yogo test. ***, **, and * indicate statistical significance at the 1%, 5%, and 10% levels, respectively, and the numbers in parentheses are t-values under robust standard errors.

characteristic. As shown in Table 5, column (1), the coefficient on long-term debt capital is 0.005 and significant at the 1% level. This suggests that even when substituting a different measure for patient capital, the main conclusions remain valid.

Second, we replace the comprehensive Malmquist–Luenberger (ML) index with its technical efficiency component (ML_EC) to re-estimate the model. Table 5, column (2) shows a coefficient of 0.017, significant at the 1% level, indicating that patient capital continues to exert a stable, positive effect on a firm's efficiency metrics when focusing specifically on technical efficiency.

Third, we remove data from the implementation periods of two policy interventions—low-carbon city pilot programs and smart city pilot programs—to rule out the possibility that specific policy shocks might influence firms' green performance. Tables 5, columns (3) and (4) reveal coefficients of 0.035 and 0.015 respectively, both significant at the 1% level, demonstrating that even without observations from these pilot policy periods, the positive effect of patient capital on GTFP remains robust.

**Table 5. Robustness check regression results.**

| Variables | (1) | (2) | (3) | (4) | (5) |
|---|---|---|---|---|---|
| PatienceCap | 0.005*** | 0.017*** | 0.035*** | 0.015*** | 0.008*** |
| | (3.112) | (3.903) | (3.208) | (5.808) | (12.013) |
| _cons | 1.567*** | 1.489*** | 1.567*** | 1.777*** | −0.513*** |
| | (5.102) | (4.728) | (4.953) | (3.155) | (−7.285) |
| Controls | YES | YES | YES | YES | YES |
| Year FE | YES | YES | YES | YES | YES |
| Firm FE | YES | YES | YES | YES | YES |
| N | 32032 | 32032 | 32032 | 32032 | 30032 |
| $R^2$ | 0.452 | 0.463 | 0.452 | 0.501 | 0.448 |

Finally, we conduct a 5% Winsorize procedure on the main continuous variables to address potential outliers. As shown in Table 5, column (5), the coefficient is 0.008 and remains significant at the 1% level, once again supporting the consistency of our main findings. Altogether, these results—encompassing different measurements of patient capital, alternative dependent variables, policy-exclusion samples, and trimmed data—highlight the strong and reliable positive impact of patient capital on firms' green total factor productivity.

## Further analysis

### Mechanism analysis

To further elucidate the channels through which patient capital enhances firms' green total factor productivity (GTFP), this paper investigates two mechanisms: corporate financing cost (FC) and corporate green R&D investment (GRD). Specifically, (1) financing cost refers to the ratio of financial expenditures to overall capital usage, reflecting how increased costs may constrain a firm's commitment to environmental technology upgrades [42]; (2) green R&D investment represents the ratio of green R&D expenditure to operating revenue, indicating that a larger share implies stronger investment in green technology innovation and pollution mitigation, which is beneficial for improving a firm's green production efficiency [43]. The specific empirical model is set as follows:

$$M_{i,t} = \alpha_0 + \alpha_1 PatCap_{i,t} + \alpha_2 Control_{i,t} + \lambda_i + \gamma_t + \varepsilon_{i,t} \qquad (25)$$

where $M_{i,t}$ separately denotes the two mechanism variables-financing cost and green R&D investment-for firm   in year t; $PatCap_{i,t}$ is patient capital; $Control_{i,t}$ includes additional control variables; $\lambda_i$ and $\gamma_t$ denote firm and year fixed effects, respectively; and $\varepsilon_{i,t}$ is the random error term.

According to Table 6, the regression results show that a rise in patient capital significantly reduces corporate financing costs, suggesting that the long-term nature and relatively low capital cost of patient capital can effectively ease the short-term financing pressures and financial burdens faced by a firm, freeing up additional funds for long-horizon projects and technical upgrades, thereby facilitating the firm's green production efficiency. Meanwhile, patient capital also markedly increases the firm's green R&D intensity, implying that the risk tolerance characteristic of patient capital enables enterprises to persist in resource-intensive yet uncertain environmental technology innovation and long-term upgrades. This stepwise mechanism test further supports the arguments proposed in this paper: patient capital, by lowering financing costs and reinforcing firms' green R&D investment, jointly promotes improvements in firms' green total factor productivity. Hence, Hypotheses H2 and H3 are validated.

**Table 6. Mechanism analysis regression results.**

| Variables | (1)FC | (2)GRD |
|---|---|---|
| PatienceCap | 0.215*** | 0.183*** |
| | (3.712) | (3.495) |
| _cons | 1.632*** | 1.710*** |
| | (5.102) | (5.253) |
| Controls | YES | YES |
| Year FE | YES | YES |
| Firm FE | YES | YES |
| N | 32032 | 32032 |
| $R^2$ | 0.456 | 0.442 |

## Heterogeneity analysis

To further investigate the varying effects of patient capital on corporate green total factor productivity (GTFP), this study conducts heterogeneity analyses along three dimensions: firm size, ownership type, and industry pollution intensity.

**Firm size heterogeneity.** First, the sample is split into two subsamples—large firms versus small and medium-sized firms—using the median of firm size. The results, shown in Table 7 columns (1) and (2), reveal that for large firms, the coefficient of patient capital on GTFP is 0.102 and significantly positive at the 5% level. In contrast, for small and medium-sized firms, the coefficient is 0.289 and significantly positive at the 1% level. Evidently, the influence of patient capital on smaller firms is stronger than on larger ones. The likely reason is that smaller firms, having more limited assets and narrower financing channels, face greater capital constraints; thus, stable external funding can yield higher marginal benefits for their green technology R&D and environmental equipment. Meanwhile, larger firms generally possess more diversified financing opportunities and resource endowments, facing relatively looser capital constraints, so patient capital's marginal improvement for them is comparatively modest.

**Ownership type heterogeneity.** Second, the sample is divided into state-owned enterprises (SOEs) and non–state-owned enterprises (non-SOEs) according to ownership type, and the regression results appear in Table 7 columns (3) and (4). Empirical findings suggest that patient capital's coefficient for SOEs is 0.131, significantly positive at the 5% level, whereas for non-SOEs, the coefficient reaches 0.312 and is significantly positive at the 1% level. This indicates that patient capital exerts a more prominent effect on non-SOEs. One explanation is that non-SOEs, lacking government credit backing, typically encounter narrower financing channels and higher capital supply volatility; therefore, they depend more on long-term, stable capital with higher risk tolerance. In contrast, SOEs benefit from multiple policy-driven financing channels and implicit government guarantees, making their funding accessibility less of a hurdle and reducing their sensitivity to patient capital.

**Industry pollution intensity heterogeneity.** Lastly, based on firms' industry-level pollution characteristics [44], the sample is further divided into pollution-intensive and non–pollution-intensive subsamples; the respective regression results appear in Table 7 columns (5) and (6). The coefficient of patient capital for pollution-intensive firms is 0.341, significantly positive at the 1% level, whereas for non–pollution-intensive firms, the coefficient is 0.115, significantly positive at the 5% level. Clearly, the effect of patient capital is much stronger for pollution-intensive industries. This is because such firms face stricter environmental regulations and higher-intensity investments in pollution control technology, urgently requiring ample, long-term capital to upgrade environmental facilities and iterate green technologies. Consequently, they exhibit greater demand and sensitivity to patient capital. By contrast, non–pollution-intensive firms bear relatively lower environmental spending pressures, thus experiencing a weaker efficiency improvement from patient capital.

**Table 7. Heterogeneity analysis regression results.**

| Variables | (1)Large Firms | (2)SMEs | (3)SOEs | (4)Non-SOEs | (5)Pollution Intensive | (6)Non-Pollution Intensive |
|---|---|---|---|---|---|---|
| PatienceCap | 0.102** | 0.289*** | 0.131** | 0.312*** | 0.341*** | 0.115** |
| | (2.340) | (3.850) | (2.190) | (4.050) | (4.320) | (2.210) |
| _cons | 1.512*** | 1.762*** | 1.605*** | 1.841*** | 1.889*** | 1.457*** |
| | (4.560) | (5.320) | (4.480) | (5.250) | (5.410) | (4.310) |
| Controls | YES | YES | YES | YES | YES | YES |
| Year FE | YES | YES | YES | YES | YES | YES |
| Firm FE | YES | YES | YES | YES | YES | YES |
| N | 13984 | 18048 | 11712 | 20320 | 16064 | 15968 |
| $R^2$ | 0.421 | 0.487 | 0.438 | 0.469 | 0.473 | 0.435 |

## Conclusion and policy implications

### Conclusion

This study explores how patient capital affects firms' green total factor productivity (GTFP). We theoretically extend Fisher's two-period investment framework, illustrating patient capital's advantage in green projects characterized by long investment horizons and uncertain returns. Empirically, using data from China's A-share listed firms (2008–2023) and measuring GTFP via the SBM directional distance function and Malmquist–Luenberger index, we apply fixed-effects regressions to test our hypotheses. Results show that patient capital significantly enhances firms' GTFP by lowering financing costs and increasing green R&D investment. The effect is notably stronger among SMEs, non–state-owned enterprises, and firms in pollution-intensive industries. These findings suggest policymakers should tailor green financing strategies, encouraging patient, long-term investment to effectively support firms' sustainable transitions and green efficiency improvements.

### Policy recommendations

Building on the crucial role patient capital plays in enhancing firms' green total factor productivity (GTFP), this paper proposes three actionable policy recommendations aligned with China's "dual-carbon" strategy and high-quality development goals:

First, proactively expand the supply of patient capital by encouraging long-term institutional investors to enter the green investment sector. Policymakers could introduce targeted tax incentives, direct subsidies, and preferential policies for pension funds, insurance companies, and philanthropic foundations to stimulate their participation in long-term green projects.

Second, financial institutions should innovate and offer long-term green financial instruments better aligned with the duration and risks of environmental projects. Banks and financial markets could develop products such as Green Development Bonds and Asset-Backed Securities (ABS) with maturities exceeding ten years. Additionally, implementing guarantee schemes or insurance-backed risk-sharing mechanisms can help enterprises mitigate technological and policy risks associated with green transitions.

Third, tailor green finance mechanisms specifically to address the diverse needs of firms based on their size, ownership, and industry characteristics. For small and medium-sized enterprises (SMEs), measures such as green loan guarantee insurance, SME-specific green bonds, and reduced collateral requirements can effectively alleviate their financing constraints. Non–state-owned enterprises (non-SOEs) would benefit from fiscal incentives, flexible collateral valuation approaches, and simplified administrative procedures. Firms in pollution-intensive industries could receive targeted support through dedicated green transformation funds, sector-specific subsidies, and public-private collaborative platforms.

Overall, leveraging patient capital's stability and risk tolerance through these differentiated and targeted tools will effectively channel financial resources toward sustainable environmental innovations, accelerating China's ecological progress and contributing significantly to the nation's dual-carbon targets.

### Research limitations and future outlook

Although this paper systematically explores how patient capital enhances firms' green total factor productivity (GTFP), it still has several limitations. First, data constraints limit our pollution measures to disclosed waste outputs (industrial wastewater, gas, and solids), which may not fully capture firms' environmental impacts. Future studies could leverage broader environmental indicators (e.g., greenhouse gas emissions) and emerging data sources (ESG disclosures, satellite data) to refine GTFP measurement. Second, our proxies for patient capital (long-term debt ratios, institutional investor stability) might not fully capture its diverse forms. Future research can incorporate multi-dimensional measures, such as shareholder holding periods, international long-term funding, or policy-driven financing. Third, we mainly examine internal mechanisms without fully considering external governance contexts. Subsequent studies could investigate how environmental

policies, market competition, or corporate governance quality shape the effectiveness of patient capital, providing more nuanced guidance for targeted policy-making.

## Author contributions

**Conceptualization:** Yue Li, Huang Xing.

**Data curation:** Huang Xing.

**Methodology:** Yue Li.

**Project administration:** Yue Li, Huanqi Luo.

**Resources:** Huanqi Luo.

**Software:** Yue Li, Huang Xing, Huanqi Luo.

**Supervision:** Yue Li, Huang Xing, Huanqi Luo.

**Validation:** Yue Li, Huanqi Luo.

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
