## [Decision Letter · Decision Letter 0]

9 Jul 2025

Dear Dr. Xing,

Thank you for submitting your manuscript to PLOS ONE. After careful consideration, we feel that it has merit but does not fully meet PLOS ONE’s publication criteria as it currently stands. Therefore, we invite you to submit a revised version of the manuscript that addresses the points raised during the review process.

We look forward to receiving your revised manuscript.

Kind regards,

Taiyi He

Academic Editor

PLOS ONE

Journal Requirements:

[This research was supported by the Ministry of Education of China Humanities and Social Sciences Youth Fund Project “Refining Local Legislation on Social Credit: An Examination of Fifteen Local Regulations” (Grant No. 21YJC820023).].

[The authors declare no conflict of interest.].

5. Please amend the manuscript submission data (via Edit Submission) to include author Luo Huanqi.

6. Please amend your authorship list in your manuscript file to include author Luo Huan qi.

Additional Editor Comments:

Please do more in-depth analysis based on the empirical results and check the details carefully.

Reviewers' comments:

Reviewer's Responses to Questions

**Comments to the Author**

1. Is the manuscript technically sound, and do the data support the conclusions?

Reviewer #1: Yes

Reviewer #2: Yes

Reviewer #3: Yes

2. Has the statistical analysis been performed appropriately and rigorously?

Reviewer #1: Yes

Reviewer #2: Yes

Reviewer #3: Yes

3. Have the authors made all data underlying the findings in their manuscript fully available?

Reviewer #1: Yes

Reviewer #2: Yes

Reviewer #3: Yes

4. Is the manuscript presented in an intelligible fashion and written in standard English?

Reviewer #1: Yes

Reviewer #2: Yes

Reviewer #3: Yes

Reviewer #1: The manuscript is well-structured and makes a useful contribution. Only minor presentation edits are required prior to acceptance. Please consider the following points.

1. The existing title is overly broad. To emphasize the study’s firm-level focus, please consider a revision such as “Patient Capital and Green Total Factor Productivity: Evidence from Chinese listed Companies.” You may adopt a different wording, but the final title should explicitly reference firms or enterprises.

2. throughout the manuscript—including main text, tables, figure captions, and footnotes—replace any full-width double quotation marks with standard half-width English double quotation marks (" ") to ensure consistent formatting.

3. Items 36–39 are not in APA format. Convert each to the journal’s required APA style.

4. The abstract should be limited to 300 words in a single paragraph, with no citations. It must briefly outline the research objectives, data and methods, key results, and policy implications, while avoiding lengthy background information or unnecessary abbreviations.

5. Replace Roman numerals (I, II, III …) in headings and sub-headings with Arabic numerals (1, 2, 3 …) to align with journal style.

Reviewer #2: The manuscript is well organized and logically cohesive, and its research content is already comprehensive. To further enhance layout and readability, I recommend only the following minor formatting and stylistic tweaks, none of which affect the study’s core conclusions.

1. The current abstract is informative but slightly wordy. Please condense it by tightening phrasing and eliminating non-essential background so key objectives, methods, and findings stand out more clearly.

2. The current description of the study’s marginal contribution is somewhat verbose. Please focus on the two or three most innovative points and shorten the section accordingly to make the argument more concise and persuasive.

3. The section outlining the study’s limitations and future research directions would benefit from greater concision. Please summarize the key points in a more streamlined paragraph.

4. Proofread for minor grammar slips (e.g., “reveal that a one–standard-deviation increase … raises,” not “raise”).

Reviewer #3: Overall, the manuscript is well organized, empirically robust, and policy-relevant, and it is already close to being publishable. Only a few minor adjustments are needed to further enhance its completeness and readability.

First, refine the introduction by pinpointing the specific research gap that the current literature has not yet addressed and by stating the core questions this study answers, avoiding overlap with existing reviews.

Second, enrich the theory section with the most recent studies on patient capital and green innovation, and end that discussion by clearly stating the incremental contribution of this paper relative to prior work.

Third, compress the abstract and conclusion so they cover only the four essentials—research purpose, data and methods, key findings, and policy implications—omitting background material that is already well explained in the text.

Fourth, add a brief institutional‑background paragraph to the theoretical framework that summarizes the evolution of China’s capital market, the green‑finance policy landscape, and the regulatory impetus of the “dual‑carbon” goals, and explain how these external conditions shape the formation and effects of patient capital.

Fifth, strengthen the explanation of the positive feedback loop between lower financing costs and green R&D: clarify how patient capital reduces refinancing risk, encourages longer‑horizon green projects, and in turn attracts additional patient capital, creating a virtuous cycle.

Sixth, make the policy section more actionable by proposing concrete tools—such as tax incentives, disclosure guidelines, and long‑term investment rewards—and tailoring recommendations to firms of different sizes and ownership types.

Seventh, integrate a corporate‑governance perspective by discussing how disclosure practices, performance metrics, and board incentives can work in tandem with patient capital to advance effective green transitions.

Eighth, standardize terminology and formatting throughout the manuscript, eliminate any Chinese–English mixtures, and ensure that all references fully comply with the journal’s style guidelines.

With these minor revisions in place, the paper should be ready for publication.

**Do you want your identity to be public for this peer review?** For information about this choice, including consent withdrawal, please see our Privacy Policy

Reviewer #1: No

Reviewer #2: No

Reviewer #3: No

---

## [Author Response · Author response to Decision Letter 1]

1 Aug 2025

Thank you very much for your thoughtful comments and constructive suggestions regarding our manuscript. We have carefully considered each comment and made detailed revisions accordingly. Specifically, we have addressed all formatting requirements, clarified funding and competing interests statements, enhanced our data availability statement, updated the authorship list, carefully reviewed and revised our references, and expanded our empirical analyses as requested.

We have provided a detailed point-by-point response in the attached document, clearly outlining how we addressed each issue raised. All changes made to the manuscript are clearly marked in bold text for your convenience.

We deeply appreciate your valuable feedback, which has significantly improved our manuscript. We hope our revisions satisfactorily address all your concerns, and we look forward to your further consideration.

---

## [Decision Letter · Decision Letter 1]

14 Aug 2025

Patient Capital and Green Total Factor Productivity: Evidence from Chinese Listed Companies

PONE-D-25-29830R1

Dear Dr. Xing,

We’re pleased to inform you that your manuscript has been judged scientifically suitable for publication and will be formally accepted for publication once it meets all outstanding technical requirements.

Kind regards,

Taiyi He

Academic Editor

PLOS ONE

Additional Editor Comments (optional):

Good jobs!

Reviewers' comments:

Reviewer's Responses to Questions

**Comments to the Author**

Reviewer #1: All comments have been addressed

Reviewer #2: All comments have been addressed

Reviewer #3: All comments have been addressed

2. Is the manuscript technically sound, and do the data support the conclusions?

Reviewer #1: Yes

Reviewer #2: Yes

Reviewer #3: Yes

3. Has the statistical analysis been performed appropriately and rigorously?

Reviewer #1: Yes

Reviewer #2: Yes

Reviewer #3: Yes

4. Have the authors made all data underlying the findings in their manuscript fully available?

Reviewer #1: Yes

Reviewer #2: Yes

Reviewer #3: Yes

5. Is the manuscript presented in an intelligible fashion and written in standard English?

Reviewer #1: Yes

Reviewer #2: Yes

Reviewer #3: Yes

Reviewer #1: The authors have made systematic and substantive revisions in response to the previous review. The literature review has been updated; the theoretical framework is more rigorous; the model specification and identification strategy are clearly articulated; robustness checks and endogeneity tests are largely complete; and the statements of conclusions and policy implications are appropriately calibrated. The major concerns raised earlier have been effectively addressed. Overall logic and writing quality have improved markedly, and the manuscript now meets the journal’s publication standards. I only suggest, at the final stage, further harmonizing terminology, standardizing figures/tables and reference formatting, and proofreading a few minor wording issues—none of which affects the conclusions or contributions. In sum, I recommend acceptance for publication.

Reviewer #2: The authors' revisions effectively address the reviewers' comments, significantly enhancing the theoretical rigor and logical clarity of the manuscript. Specifically, notable improvements have been made in clearly defining the research questions and articulating the theoretical mechanisms, resulting in a more coherent and internally consistent structure. Additionally, the newly added robustness tests effectively strengthen the reliability of the study’s conclusions, and the policy implications section now provides more concise and targeted recommendations. Overall, this revision adequately resolves the issues raised previously, and I recommend the manuscript for acceptance.

Reviewer #3: The revised manuscript is substantively improved and, in my view, ready for publication: the research question is now sharply defined; the theoretical framework is coherent with clearly testable hypotheses; the identification strategy is transparent with well-documented sample and variable construction; and the added robustness exercises (parallel-trend diagnostics, placebo tests, alternative measures, and sample restrictions) consistently support the main findings. Potential endogeneity concerns are addressed with appropriate treatments yielding stable results, while the mechanism and heterogeneity analyses are focused and aligned with the theory. The abstract, conclusions, and policy implications are concise and consistent with the evidence, and data-availability and reference formatting are largely compliant. Only minor editorial issues remain (e.g., harmonizing axis labels/decimal places and a final check of reference page numbers/DOIs), which can be handled at proof stage. I recommend acceptance without further substantive revision.

**Do you want your identity to be public for this peer review?** For information about this choice, including consent withdrawal, please see our Privacy Policy

Reviewer #1: No

Reviewer #2: No

Reviewer #3: No

---

## [Editor Report · Acceptance letter]

PONE-D-25-29830R1

PLOS ONE

Dear Dr. Xing,

I'm pleased to inform you that your manuscript has been deemed suitable for publication in PLOS ONE. Congratulations! Your manuscript is now being handed over to our production team.

Kind regards,

on behalf of

Dr. Taiyi He

Academic Editor

PLOS ONE